# Effects of Topical 1,25 and 24,25 Vitamin D on Diabetic, Vitamin D Deficient and Vitamin D Receptor Knockout Mouse Corneal Wound Healing

**DOI:** 10.3390/biom13071065

**Published:** 2023-07-01

**Authors:** Xiaowen Lu, Zhong Chen, Jerry Lu, Mitchell Watsky

**Affiliations:** Department of Cellular Biology and Anatomy, Medical College of Georgia at Augusta University, Augusta, GA 30912, USA

**Keywords:** corneal wound healing, 1,25-dihyroxy vitamin D, 24,25-dihydroxy vitamin D, Vit D receptor

## Abstract

Delayed or prolonged corneal wound healing and non-healing corneas put patients at risk for ocular surface infections and subsequent stromal opacification, resulting in discomfort or visual loss. It is important to enhance corneal wound healing efficiency and quality. Vitamin D (Vit D) is both a hormone and a vitamin, and its insufficiency has been linked to immune disorders and diabetes. For this study, wound healing and recruitment of CD45+ cells into the wound area of normoglycemic and diabetic mice were examined following corneal epithelial debridement and treatment with 1,25-dihyroxyvitamin D (1,25 Vit D) or 24,25-dihydroxyvitamin D (24,25 Vit D). Treatment with topical 1,25-dihyroxyvitamin D (1,25 Vit D) resulted in significantly increased corneal wound healing rates of normoglycemic, diabetic and diabetic Vit D deficient mice. Furthermore, 24,25-dihydroxyvitamin D (24,25 Vit D) significantly increased corneal wound healing of diabetic Vit D deficient and Vit D receptor knockout (VDR KO) mice. In addition, CD45+ cell numbers were reduced in diabetic and VDR KO mouse corneas compared to normoglycemic mice, and 24,25 Vit D increased the recruitment of CD45+ cells to diabetic mouse corneas after epithelial debridement. CD45+ cells were found to infiltrate into the corneal basal epithelial layer after corneal epithelial debridement. Our data indicate that topical Vit D promotes corneal wound healing and further supports previous work that the Vit D corneal wound healing effect is not totally VDR-dependent.

## 1. Introduction

The cornea is the primary refractive element of the eye and acts as the eye’s barrier to the external environment, protecting underlying ocular tissues from infection and injury. Over 10 million people worldwide are currently affected by corneal blindness [1]. Diabetes mellitus is the most common cause of blindness in working age populations [2]. Diabetic keratopathy is the most frequent clinical condition affecting the cornea in diabetics, occurring in up to 70% of all diabetics [3,4,5]. Diabetic keratopathy can exhibit several clinical manifestations, including (but not limited to) persistent corneal epithelial erosion, delayed epithelial regeneration, an altered ocular surface immune response, increased corneal thickness, altered stromal crosslinking and biomechanical properties and decreased corneal sensitivity [2,5]. These features are typically caused by changes within the corneal epithelial basement membrane, deposition of glycation products, corneal nerve ending damage, reduced tear secretion and oxidative stress related to the hyperglycemic condition [5]. Delayed or prolonged corneal wound healing or non-healing puts patients at risk for ocular surface infections, surface irregularities and stromal opacification, resulting in discomfort or visual loss [3]. It is important to enhance corneal wound healing efficiency and quality in healthy and diabetic patients [6]. Although significant progress has been made in recent years, pharmaco-therapeutic agents that promote corneal healing remain extremely limited [7].

It is estimated that 1 billion people worldwide are vitamin D (Vit D) insufficient or deficient [8]. Vit D insufficiency has been linked to numerous pathological conditions including immune/autoimmune disorders and diabetes [9,10,11,12]. Vit D supplementation may diminish the prevalence and adverse outcomes of these diseases and may reduce all-cause mortality [13,14,15]. Patients with hypovitaminosis D have been shown to have significantly altered corneal endothelial parameters including cell density and hexagon cell ratio [16]. In addition, Shetty et al. correlated low serum Vit D levels with evaporative dry eye disease [17] and also reported a case in which there was dramatic improvement following Vit D replacement in a Vit D deficient patient presenting with bilateral Meibomian gland dysfunction and evaporative dry eye symptoms [18].

Vit D has been shown to contribute to corneal wound healing and epithelial integrity and to upregulate the expression of tight junctions and structural proteins [19,20]. Our group has demonstrated that Vit D deficient (VDD) diabetic mice have slower cornea epithelial wound healing than wildtype (WT) diabetic mice. We also found that Vit D receptor knockout (VDR KO) mice have significantly slower corneal wound healing rates and reduced corneal nerve densities [21]. Moreover, reduced wound healing rates and nerve densities were not rescued by a supplemental diet rich in calcium, lactose and phosphate, demonstrating that these effects were not due to Vit D deficiency-related hypocalcemia [22]. Our group also demonstrated significant roles for Vit D in maintaining corneal epithelial adherens junctions, promoting gap junction communication and in positively affecting cell migration and proliferation [23,24]. To date, only a handful of studies have directly explored the therapeutic potential of using topical Vit D for corneal epithelial wound healing, corneal neovascularization or bacterial-induced inflammatory cytokine production [25,26,27], and none of these studies examined the effects of topical 24,25-dihydroxyvitamin D3 (24,25 Vit D) on corneal wound healing. This current study focused on the influence of topical 1,25-dihyroxyvitamin D (1,25 Vit D) and 24,25 Vit D on corneal wound healing in normoglycemic, diabetic, VDR KO and VDD mice.

Infiltrating immune cells are recruited to the cornea following injury. Neutrophils are the dominant infiltrating cells in the early stage of inflammation. Corneal neutrophils have been found to contribute to corneal re-epithelialization and angiogenesis [28,29], and corneal wound healing has been found to be delayed in neutropenic WT mice [30]. Vit D has been shown to significantly increase infiltration of neutrophils into the central cornea stroma [31]. The current study examines how Vit D influences corneal wound healing and affects CD45+ cell infiltration into the corneal stroma and epithelium following epithelial abrasion. It also explores the potential association between the dysfunctional wound healing response, Vit D and altered inflammation in diabetic mice.

## 2. Materials and Methods

### 2.1. Materials

For this study, 1,25 Vit D and 24,25 Vit D were purchased from Enzo Life Sciences (Catalog# BML-DM200-0050, BML-DM300-0050, Farmingdale, NY, USA). Antibodies for anti-CD45 were purchased from ABCAM (Catalog# ab10558, Cambridge, MA, USA). Mouse monoclonal pan-cytokeratin (PanCK) antibody was purchased from Santa Cruz Biotechnology Inc. (Catalog # SC-81714, Santa Cruz, CA, USA).

### 2.2. Animal Experiments

All animal studies were approved by the University IACUC, and animals were treated according to the ARVO statement for the Use of Animals in Ophthalmic and Visual Research. Animals were housed in standard conditions with a 12 h dark–light cycle. If not otherwise stated, at the end of the experiments, the mice were killed with CO_2_ inhalation and neck dislocation, and tissues were collected. The low-dose streptozotocin (STZ) injection method was used to induce diabetes. Briefly, five sequential daily intraperitoneal injections of a freshly prepared solution of STZ in 0.1 mol/L citrate buffer (pH 4.5) at a dosage of 60 mg/kg body weight were administered to 4-week-old mice. Blood glucose was measured 1 week after the final STZ injection. We limited the mice receiving corneal epithelial wounds to only those with blood glucose levels >249 and <651 mg/dL [32]. Breeding pairs of VDR KO mice which displayed a range of traits including hypocalcemia, abnormal blood mineral levels and growth retardation were purchased from Jackson Labs (strain: B6.129S4-Vdr<tm1Mbd>/J). Mice were bred in-house, and males and females were used for all studies.

### 2.3. Animal Groups

C57BL/6 mice were used in our studies, including WT (CTRL), diabetic (Diab), diabetic Vit D deficient mice (Diab VDD), VDR KO and diabetic VDR knockout mice (Diab VDR KO). As previously described and characterized [22], VDD was induced by feeding mice a Vit D deficient diet (TD.89123 Diet; Envigo) for 4 weeks. VDD control mice were fed TD.89124 vitamin D control diet (Envigo). The duration of diabetes in the Diab groups was 10 weeks.

### 2.4. Mouse Corneal Wounding and Topical Vit D Administration

After anesthesia with isoflurane (Tec 4 vaporizer, ASE Anesthesia Service and Equipment, Atlanta, GA, USA), a 2 mm central corneal epithelial wound was made with an Algerbrush II (Ambler Surgical, Exton, PA, USA). To monitor wound size, a drop of fluorescein sodium 0.25% with proparacaine hydrochloride 0.5% was applied to the wounded eye. Antibiotic ointment (bacitracin, neomycin, polymixin) and 0.05% buprenorphine (intramuscular) were administered after every procedure. Mouse corneas were treated with 40 nM 1,25 Vit D, 200 nM 24,25 Vit D or vehicle control, separately. Vit D was topically administered (one drop, three times per day) starting immediately after corneal epithelial debridement. Wounds were photographed with a Topcon slit lamp (SL-D4) at 0, 6, 18, 24, 28 h or until the wounds were healed.

### 2.5. Corneal Whole-Mount Confocal Imaging and Three-Dimensional Reconstruction

CD45+ immune cell (marker of hemopoietic origin) infiltration was measured and imaged following corneal wounding. A 2 mm corneal epithelial debridement or a single superficial scratch wound approximately 0.1 mm in length was made on the corneal surface of 8 week old C57BL/6 mice to stimulate CD45+ cell recruitment into the corneas. Debridement wounds were made for cell number quantification, while the scratch wound was made to allow for 3D reconstructions of immune cell infiltration (see below). Whole eyes were excised 8 h after the scratch wound and 24 h after debridement wounding and fixed with Zamboni fixative (American MasterTech Scientific, Lodi, CA, USA) for 75 min, then washed with PBS (three times). Corneas were carefully excised along the sclera corneal rim. To block nonspecific binding, corneas were incubated with 10% normal goat serum plus 0.1% Triton X-100 solution in PBS for 60 min at room temperature. Corneas were incubated with primary anti-CD45 (1:200) and anti-PanCK (1:200) antibody in PBS containing 5% goat serum plus 0.1% Triton X-100 for 24 h at 4 °C and constantly shaken. After washing with PBS three times for 10 min each, corneas were incubated with (1:1000) Alexa Fluor 488 goat anti-rabbit IgG (H + L) and Alexa fluor plus 555 goat anti-mouse IgG (A-32731, A-21422, Thermo Fisher Scientific, Norcross, GA, USA) secondary antibodies for 24 h at 4 °C and washed thoroughly with PBS. DAPI was used to label nuclei, allowing for differentiation of the different cornea layers. Confocal analysis was performed using a Zeiss LSM 780 laser-scanning confocal microscope with a 40× objective. Z-stacks were generated in 1 μm step increments, and 3D reconstructions were performed using ZEN lite software.

### 2.6. Recruitment of CD45 Cells in Wounded WT and VDR KO Mouse Corneas

To detect the effects of VDR KO on the recruitment of CD45+ cells, a 2 mm central epithelial wound was made using an Algerbrush 2 in 8-week-old normoglycemic C57BL/6 mice and 8-week-old VDR KO mice. After 8 h, eyes were enucleated and processed as previously described using the corneal whole-mount process. Tissues were then incubated with primary anti-CD45 (1:200) and secondary antibodies (1:1000) Alexa Fluor 488 goat anti-rabbit IgG. CD45+ cells in the anterior portion of the stroma were imaged with a Leica Stellaris Confocal Microscope (Danaher Corporation, Washington, DC, USA).

CD45+ cell numbers were quantified using the Imaris image processing software Spots package (Oxford Instruments, Abingdon, UK). To quantify the number of CD45+ cells, the “Spots” function was used. While Imaris is inherently a 3D/4D visualization software, this specific function also worked well to count individual cells within 2D images. After inputting a 2D image into Imaris (via selecting a negligible z voxel), steps were taken to calibrate the spots function. By selecting the average diameter of the cells and setting a fixed initial sensitivity, the software labels individual cells as spots and reports a cumulative spots count for each image. After an initial rough pass in which the count was completely automated by the software, overexposed and underexposed areas were accounted for by manual spot deletion or addition. A conservative approach was taken such that only clear cells (spots) in heavily saturated parts of the image were counted.

### 2.7. Recruitment of CD45 Cells in Wounded Diabetic Mouse Corneas Treated with Vit D

To detect the effects of topical Vit D on the recruitment of CD45+ cells, a 2 mm central epithelial wound was made using an Algerbrush 2 in normoglycemic and 12-week-old diabetic (8 week diabetes duration) and normoglycemic C57BL/6 mice. The mouse corneas were treated with 40 nM 1,25 Vit D, 200 nM 24,25 Vit D or vehicle control. Vit D was topically administered as one drop, four times over 8 h, starting immediately after the epithelial debridement. After 8 h, eyes were enucleated and processed as previously described using the corneal whole-mount process. The tissues were then incubated with primary anti-CD45 (1:200) and secondary antibodies (1:1000). CD45+ cells in the anterior portion of the stroma were imaged and counted as described above.

### 2.8. Statistical Analysis

Comparisons were made between groups using Student’s T-tests. The effects of Vit D on CD45+ cell numbers in the anterior stroma were analyzed with ANOVA (GraphPad Software, La Jolla, CA, USA).

## 3. Results

### 3.1. Normoglycemic Mouse Corneal Wound Healing

Figure 1a shows representative wound healing photos of vehicle-treated and 1,25 Vit D-treated WT mouse corneas. Our results show that 1,25 Vit D-treated mice typically healed completely in less than 28 h (26 ± 2.3 h, *n* = 5), while vehicle-treated mice were often not completely healed 30 h after surgery (30 ± 1.6 h, *n* = 5). Total healing times were significantly different between vehicle-treated and 1,25 Vit D-treated mice (Figure 1b). Wounded WT mice were also treated with 200 nM 24,25 Vit D, and while the mean total healing time was lower in the 24,25 Vit D-treated mice compared to WTs (28 ± 1.6 h, *n* = 5), the difference was not significant.

### 3.2. Diabetic Mouse Corneal Wound Healing

Figure 2a shows representative wound healing photos of vehicle and 1,25 Vit D-treated diabetic mouse corneas. Vit D-treated diabetic corneas typically healed in approximately 40 h (41 ± 5.1 h, *n* = 5), while vehicle-treated diabetic mouse corneas were often not completely healed 50 h after surgery (51 ± 4.3 h, *n* = 5) (Figure 2b). Total healing times were significantly different between vehicle-treated diabetic and 1,25 Vit D-treated diabetic mice. Wounded diabetic mice were also treated with 200 nM 24,25 Vit D, and similar to the normoglycemic mice, mean total healing time was lower in the 24,25 Vit D-treated mice (47.6 ± 1.7 h, *n* = 5), but that difference was not significant.

### 3.3. Diabetic VDD Mouse Corneal Wound Healing

Figure 3 shows the total healing times of vehicle-treated and Vit D-treated diabetic VDD mice. Treated mice typically healed in approximately 50 h (49.7 ± 4.8 h for 1,25 Vit D, 51 ± 5.7 h for 24,25 Vit D, *n* = 5), while vehicle-treated diabetic mice were often not completely healed 66 h after surgery (65.5 ± 4.3 h, *n* = 5). Total healing times were significantly different between untreated diabetic VDD mice and both 1,25 and 24,25 Vit D-treated diabetic VDD mice.

### 3.4. VDR KO Mouse Corneal Wound Healing

Figure 4a shows the total healing times of untreated and 24,25 Vit D-treated VDR KO mice. Because VDR was not active, and 1,25 D is the primary agonist of VDR, VDR KO mice were not treated with 1,25 D. Furthermore, 24,25 Vit D-treated mice typically healed in approximately 31 h (31.2 ± 6.6 h, *n* = 5), while vehicle-treated VDR KO mice were often not completely healed 38 h after surgery (38.4 ± 3.3 h, *n* = 5) (Figure 4b). Total healing times were significantly different between untreated and 24,25 Vit D-treated VDR KO mice.

### 3.5. Distribution of CD45+ Cells in Control and Algerbrush-Wounded Normoglycemic Mouse Corneas

In the periphery of unwounded corneas, there was a sporadic distribution of CD45+ cells in the upper 1/3 of the stroma, with a few CD45+ cells also seen in the deep stromal layer close to endothelium (Figure 5a). Examining CD45+ cells in the peripheral cornea 8 h after the central epithelial injury, several CD45+ cells were observed within the basal epithelium. Many CD45+ cells were localized within the anterior stroma and in the posterior stroma close to the endothelium. Not as many cells were observed in the mid-stroma. Additionally, 24 h after epithelial injury, CD45+ cells were distributed throughout all depths of the peripheral cornea stroma.

Examining the center of unwounded corneas (Figure 5b), a small number CD45+ cells were observed in the anterior stroma, with few CD45+ cells observed in the deeper stromal layers. In the central cornea 8 h after injury, there was a small number of CD45+ cells in the basal layer of epithelium at the wound edge. Compared to unwounded corneas, there were fewer CD45+ cells in the anterior stroma, and only a few CD45+ deeper layers of the stroma. In addition, 24 h after injury, CD45+ cell distribution appeared similar to that of unwounded corneas.

### 3.6. Distribution of CD45+ Cells in the Wounded Area of Normoglycemic Mice: 3D Reconstruction

It is difficult to reconstruct and analyze 3D views of corneas with relatively large circular central wounds. To allow for 3D reconstructions of CD45+ cells moving into the cornea following wounding, a 0.1 mm scratch wound was made in normoglycemic mouse corneas. In addition to labeling CD45+ cells, DAPI was used to label nuclei and PanCK to label epithelial cells. Reconstructed images were virtually rotated, allowing for a comprehensive 3D visualization of the corneal microenvironment. Figure 6 shows a representative 3D reconstruction of an immunostained cornea. CD45+ cells were detected within the basal epithelial cell layer 8 h after injury. Figure 7 demonstrates this in similarly labeled whole mount corneas 24 h after a debridement injury.

### 3.7. Effects of VDR KO on Recruitment of CD45+ Cells following Wounding

Confocal images of whole-mount VDR KO corneas 8 h after a 2 mm diameter corneal epithelial debridement are shown in Figure 8. Quantification of CD45+ cells demonstrates reduced CD45+ cell numbers in VDR KO (8110.7 ± 792.2, *n* = 3) vs. WT corneas (12,478.3 ± 2121.5, *n* = 3).

### 3.8. Effects of 1,25 and 24,25 Vit D on Recruitment of CD45+ Cells following Diabetic Mouse Corneal Wounding

CD45+ cells were detected in the anterior stroma 8 h after epithelial abrasion. The number of CD45+ cells was significantly decreased in untreated diabetic (6331.7 ± 1496.6, *n* = 3) compared to normoglycemic mice (12,478.3 ± 2121.5, *n* = 3) (Figure 9A,B). The number of CD45+ cells was significantly increased in diabetic mice treated with topical 24,25 Vit D (9203.7 ± 931.3, *n* = 3) (Figure 9D,E), while there was no significant difference in the number of CD45+ cells in diabetic mice treated with topical 1,25 Vit D (7570.5 ± 426.3, *n* = 3) as compared to untreated diabetic mice.

## 4. Discussion

Vit D is both a hormone, as it can be synthesized from 7-dehydrocholestrol, and a vitamin, because most populations do not synthesize enough Vit D by the de novo pathway. It was found that higher serum Vit D levels have a favorable effect on dry eye syndrome symptoms [33], while dry eye and impaired tear function in patients have been correlated with Vit D deficiency [34,35,36]. In line with these findings, Shin et al. recently reported that Vit D status affects the efficacy of topical artificial tears in dry eye patients, and Vit D supplementation improved the efficacy [37]. Alsalem et al.’s characterization of Vit D in ocular barrier cells supports the hypothesis introduced by our group that Vit D has a role in barrier function and immune regulation in ocular barrier epithelial cells [19,21,38]. Reins et al. reported that topical 1,25 Vit D treatment (100 nM) of mouse corneas resulted in a significantly delayed rate of wound closure after both 12 and 18 h, although there was no significant difference in the total wound healing time when compared to vehicle treatment [31]. Jabbehdari et al. reported that 100 nm 1,25 Vit D induced faster scratch wound closure compared with 1000 nm and 10 nm 1,25 Vit D treatment in normoglycemic mice, and a single treatment with 100 nM 1,25 Vit D significantly improved corneal wound healing compared to multiple treatments and vehicle treatment [20]. Wang et al. reported that 1 nM 1,25 Vit D promoted diabetic mouse corneal wound healing [39]. The current study demonstrated that topical 40 nM 1,25 Vit D significantly enhanced the corneal wound healing process of normal WT and diabetic mice. These studies make it increasingly clear that Vit D plays a beneficial role in the eye and particularly in the anterior segment.

In the Vit D metabolic pathway, both 25(OH)D and 1,25 Vit D can be acted on by 24-hydoxylase to form 24,25 Vit D. It has only recently been recognized that 24,25 Vit D is more than a metabolic byproduct of Vit D metabolism, demonstrating modest beneficial effects on bone [40]. Furthermore, 24,25 Vit D can exert its actions through the VDR receptor, although with a binding affinity 100 times less than 1,25 Vit D [41]. Our group previously demonstrated that 24,25 Vit D is the most prevalent Vit D metabolite in the rabbit eye [42]. We also determined that 24,25 Vit D participates in the corneal epithelial cell Vit D metabolite feedback pathways and stimulates corneal epithelial cell migration and proliferation (which is inhibited by 1,25 Vit D) [43]. In addition, 24,25 Vit D stimulates gap junction protein expression and connectivity, as well as epithelial desmosome and hemidesmosome protein expression [24,44]. Relevant to the current study, we also have demonstrated that 24,25 Vit D (and possibly 1,25 Vit D) can likely exert their actions in the cornea via a receptor distinct from VDR [24,43,45]. Supporting this hypothesis, the current study demonstrated that 24,25 Vit D increases corneal wound healing times in diabetic VDD and VDR KO mice.

In the normal human and mouse cornea, these are a significant number of CD45+ cells within the pericentral and central region [46,47,48]. Additionally, 50% of CD45+ cells co-expressed macrophage markers (F4/80), and a small to negligible numbers of cells expressed dendritic cell (CD11c) or granulocyte (Ly6G) markers [48]. For injured mouse corneas, Reins et al. found that topical 1,25 Vit D administration acutely increased granulocyte (Ly6G) infiltration in the mouse stroma [49]. After a scrape wound, neutrophils extravasate from the limbal vessels and migrate through the corneal stroma to the site of damage [50,51]. Neutrophil recruitment begins within 15 min and reaches a plateau level by 4–8 h in mice [28], with neutrophil numbers returning to baseline levels around 48 h [28]. Rapid mobilization of inflammatory cells within hours after a distant corneal injury suggests that these cells are likely involved in the acute inflammatory response, which has been shown to be critical for efficient wound healing in the cornea. Despite an increased awareness of the presence of corneal inflammatory cells in recent years, data demonstrating their precise role in normal corneal homeostasis are rare [52].

Most human and mouse studies examining infiltrating immune cells in the cornea have focused on recruitment of the cells to the stroma following corneal injury and the function of the corneal inflammatory responses. Previous studies found that most CD45+ cells resemble macrophages [48,53,54,55]. It is possible that these macrophages have a multipotent capacity to differentiate into keratocytes and contribute to collagen synthesis following injury, as has been demonstrated in vitro using primary cultures of isolated human corneal stromal cells [56]. Although resident immune cells distributed throughout the corneal epithelium and stroma include dendritic cells, macrophages, mast cells and innate lymphoid cells [57,58], there are no studies noting neutrophils infiltrating from the stroma into the corneal epithelium. In the current study, the ability to virtually rotate the 3D reconstructed images of whole wounded corneas allowed for visualization of the cornea from different angles and provided a unique and representative image of the interactions of migratory inflammatory cells and the stromal and epithelial microenvironment. A significant number CD45+ cells were detected in the basal epithelial layer at 8 h after epithelial debridement, with CD45+ cells distributed throughout the basal epithelial layer and stroma at 24 h. A significant number of CD45+ were also located along the basement membrane at the wounded edge. It has been noted that the human corneal stroma harbors CD133+/CD45+/CD14+ monocytic progenitor cells [56]. We hypothesize that in addition to their immune function, these neutrophils are involved in re-epithelialization of the wounded cornea.

VDR is expressed in monocytes and activated T and B cells [59,60]. A recent study determined that monocytes, macrophages, neutrophils, and hematopoietic stem cells are mostly derived from VDR positive lineages [61]. VDR KO mice display important defects in macrophage function and cellular immunity in vitro and in vivo. Macrophage chemotaxis is impaired in VDR KO mice [62]. VDR KO mice also have fewer tolerogenic dendritic cells in the gut [63]. In the current study, CD45+ cells were found to be significantly decreased in wounded VDR KO mice as compared to WT normoglycemic mice. The reduced infiltration of CD45+ cells into wounded VDR KO mouse corneas may be directly related to the lack of VDR in these cells or a change in inflammatory signaling in these corneas.

Diabetes is known to increase susceptibility to infections, partly due to changes in innate immunity [64]. Neutrophil migration is reduced in septic diabetic mice and humans [65,66]. Injured diabetic mouse corneas have been reported to have an impaired ocular surface adaptive immune response compared to normoglycemic mice, along with an impaired systemic adaptive immune response [67]. This is consistent with our finding of reduced CD45+ cell numbers in wounded diabetic mouse corneas compared to normoglycemic mice. While it is well known that Vit D can regulate the immune system to facilitate prevention of infections [68], there have been conflicting results concerning the effects of Vit D on the diabetic corneal immune cell. Wang et al. reported that impaired diabetic corneal wound healing was accompanied by excessive neutrophil infiltration, with topical 1,25 Vit D application efficiently reducing the number of infiltrating neutrophils in diabetic mice [39]. However, Reins et al. reported that Vit D possesses the protective ability to increase innate corneal immunity-related proteins and stimulate neutrophil infiltration [31]. In the current study, CD45+ cells were found to be significantly decreased in diabetic mice as compared to normoglycemic mice, with 24,25 Vit D but not 1,25 Vit D, which significantly increased CD45+ cells in wounded diabetic mouse corneas. A limitation to the current study is that immune cells were only immunolabeled with an anti-CD45 antibody and not immunophenotyped by flow cytometry.

## 5. Conclusions

Topical 1,25 Vit D application significantly accelerated corneal wound healing of normoglycemic, diabetic and diabetic VDD mice. Topical 24,25 Vit D significantly accelerated corneal wound healing of diabetic VDD and VDR KO mice. After corneal epithelial debridement, CD45+ cells infiltrated into the stroma and basal corneal epithelial layer. The number of CD45+ cells was reduced in diabetic and VDR KO mice after wounding as compared to normoglycemic mice. Topical 24,25 Vit D, but not 1,25 Vit D, increased the recruitment of CD45+ cells in diabetic mouse corneas following corneal epithelial debridement. These results indicate that topical 1,25 Vit D and potentially topical 24,24 Vit D may be useful for treating cornea epithelial wounds, particularly in Vit D deficient and diabetic patients. Reduced CD45+ cell recruitment in diabetic and VDR KO mouse corneas, which typically have worse epithelial wound healing outcomes compared to normal corneas, would indicate that CD45+ cell recruitment, at least in moderation, is beneficial for epithelial wound healing. Given the increased CD45+ recruitment in wounded diabetic corneas treated with 24,25 Vit D, this might be a preferable treatment for wounded diabetic corneas. The short half-life of 1,25 D and high cost of both 1,25 and 24,25 D3 may make cholecalciferol (Vit D3), which can be converted to both 1,25 D3 and 24,25 D3 in the cornea [42,69], a more reasonable choice for topical treatment of corneal wounds in diabetic and vitamin D deficient patients.

## Figures and Tables

**Figure 1 biomolecules-13-01065-f001:**
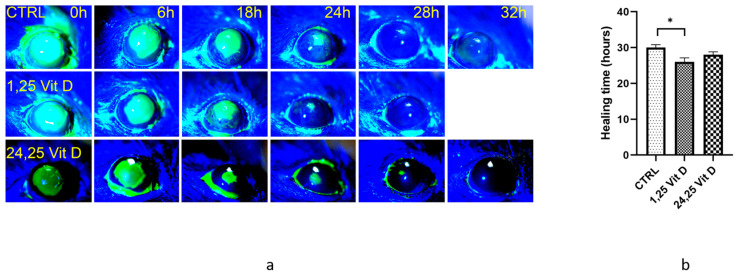
1,25 Vit D significantly accelerated corneal wound healing in normoglycemic mice. (**a**) Representative slit lamp images of mouse corneal wound healing following treatment with 1,25 Vit D. (**b**) Time until cornea wound closure following topical 1,25 Vit D treatment (* *p* < 0.05 compared to control, *n* = 5).

**Figure 2 biomolecules-13-01065-f002:**
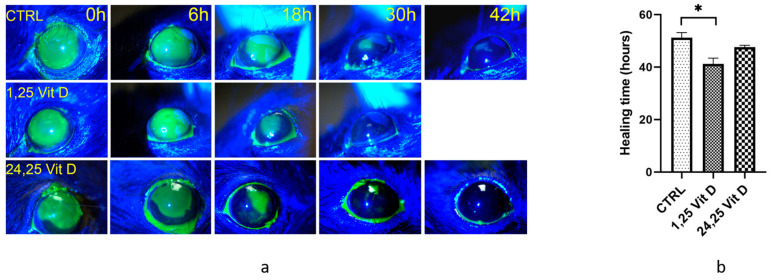
1,25 Vit D significantly accelerated corneal wound healing of diabetic mice, while 24,25 Vit D had no effect. (**a**) Representative slit lamp images of diabetic mouse corneal wound healing following topical 1,25 Vit D treatment. (**b**) Time until wound closure for diabetic mice treated with 1,25 Vit D and 24,25 Vit D (* *p* < 0.05 compared to control, *n* = 5).

**Figure 3 biomolecules-13-01065-f003:**
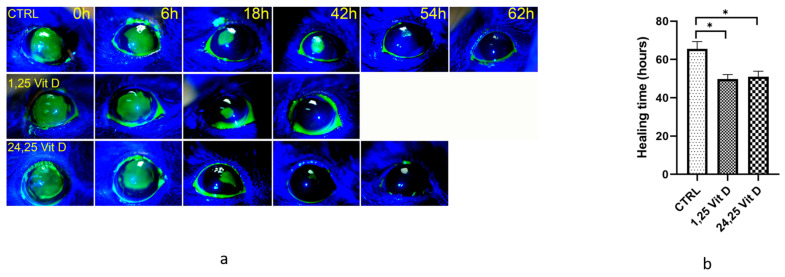
1,25-Vit D and 24,25-Vit D both significantly accelerated corneal wound healing of diabetic VDD mice. (**a**) Representative slit lamp images of diabetic VDD mouse corneal wound healing following topical 1,25 Vit D and 24,25 Vit D treatment. (**b**) Time until wound closure for diabetic VDD mice treated with 1,25 Vit D and 24,25 Vit D (* *p* < 0.05 compared to control, *n* = 5).

**Figure 4 biomolecules-13-01065-f004:**
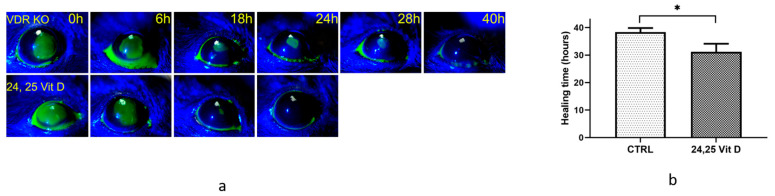
24,25-Vit D significantly accelerates corneal wound healing in VDR KO mice. (**a**) Representative slit lamp images of VDR KO mouse corneal wound healing following topical 24,25 Vit D treatment. (**b**) Time until wound closure for VDR KO mice treated with 24,25 Vit D (* *p* < 0.05, *n* = 5).

**Figure 5 biomolecules-13-01065-f005:**
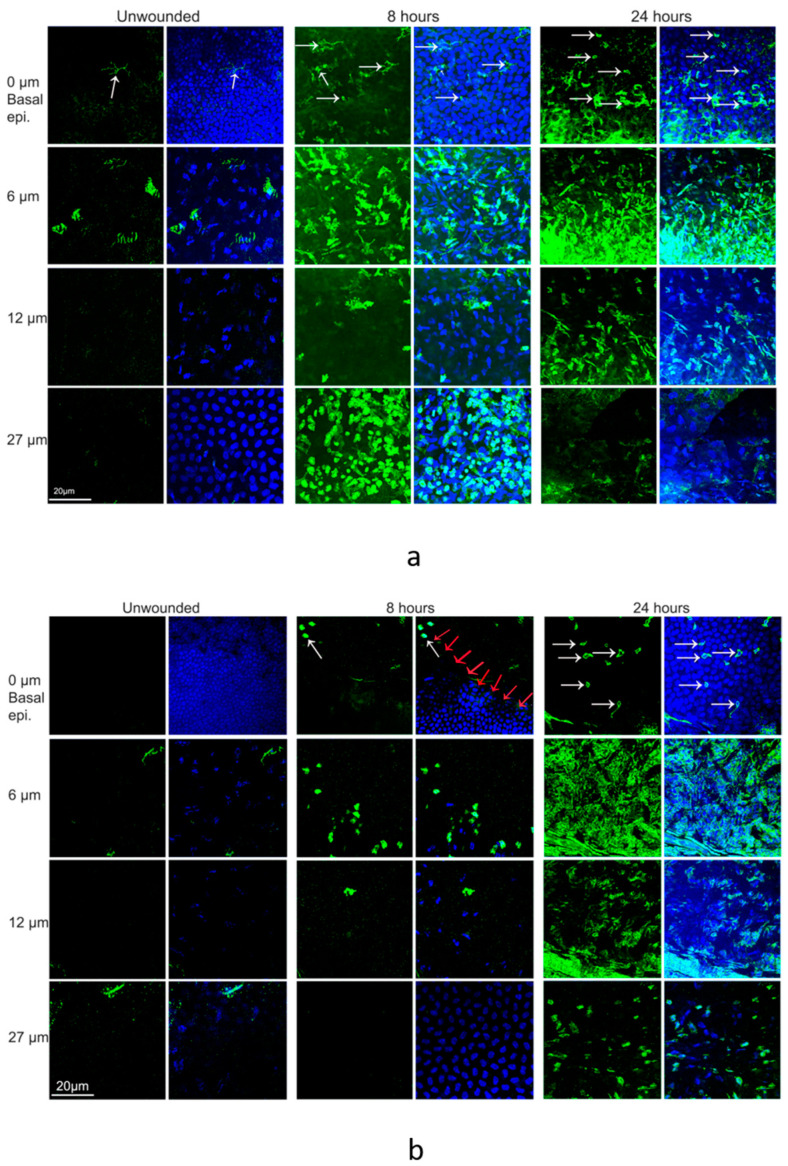
Distribution of CD45+ cells in unwounded and wounded normoglycemic mice at 8 h and 24 h. Corneas from wounded mice were harvested at 8 h and 24 h for immunohistochemistry. Corneas were immunostained with an anti-CD45 antibody (green) and DAPI (blue). (**a**) Representative images illustrating the distribution of CD45+ cells from the basal epithelia to endothelium in the peripheral cornea. Depth of the corneal section is shown on the left. (**b**) Distribution of CD45+ cells from the basal epithelia to endothelium in the central cornea (red arrows point to the wound edge, and white arrows point to CD45+ cells in the sub-basal epithelium).

**Figure 6 biomolecules-13-01065-f006:**
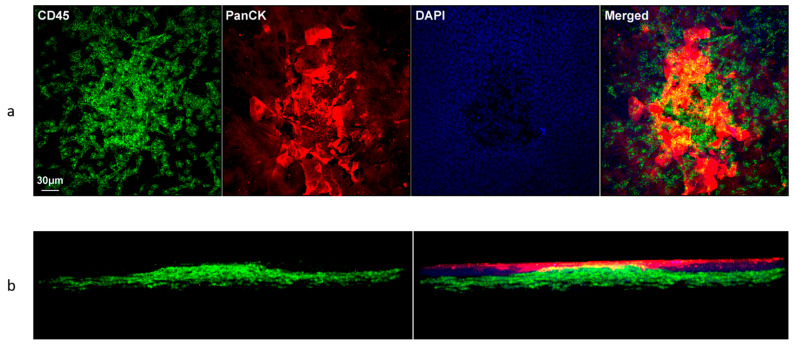
Recruitment of CD45+ cells in wounded epithelium following a scratch wound in normoglycemic mice. Corneas from wounded mice were harvested 8 h after wounding for immunohistochemistry. Corneas were immunostained with anti-CD45 (green) and DAPI (blue) then merged. (**a**) Representative top-down reconstructed three-dimensional image of a wounded cornea demonstrates the distribution of CD45+ cells. (**b**) Rotated side angle view of the cornea in (**a**) demonstrating the distribution of CD45+ cells through the anterior stroma and basal epithelium.

**Figure 7 biomolecules-13-01065-f007:**
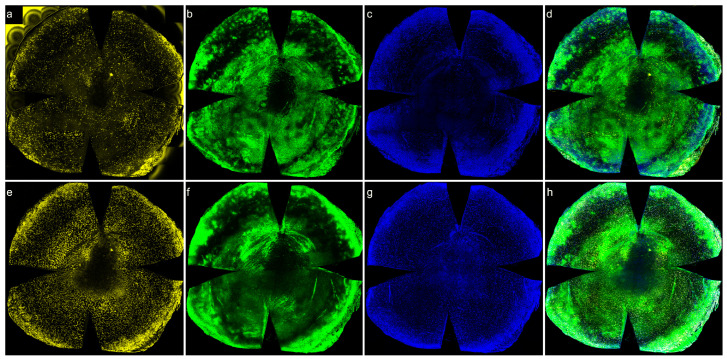
Recruitment of CD45+ cells in wounded cornea epithelium following a 2 mm epithelial debridement in normoglycemic mice. Corneas from wounded mice were harvested after 24 h for immunohistochemistry. Corneas were immunostained with anti-CD45 (yellow), anti-PanCK (green) antibodies and DAPI (blue) then merged. (**a**–**d**) Representative whole-mount corneas illustrating CD45+ and PanCK (epithelia cells) immunostaining in the sub-basal epithelium of wounded normoglycemic mouse corneas. (**e**–**h**) CD45+ and PanCK immunostaining in the anterior stroma of wounded normoglycemic mouse corneas.

**Figure 8 biomolecules-13-01065-f008:**
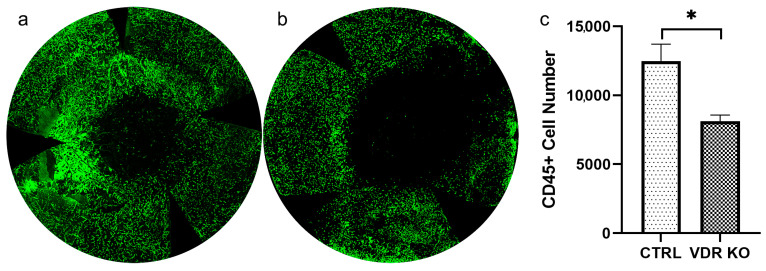
CD45+ cells in the anterior stroma of wounded WT and VDR KO mice. Corneas were wounded with a 2 mm corneal epithelial debridement, and corneas were collected and immunostained with an anti-CD45 (green) antibody 8 h after corneal abrasion in (**a**) WT and (**b**) VDR KO mice. (**c**) CD45+ numbers were quantified using the Imaris image processing software Spots package, demonstrating elevated CD45+ immune cell numbers in WT vs. VDR KO corneas (*n* = 3, * *p* < 0.05).

**Figure 9 biomolecules-13-01065-f009:**
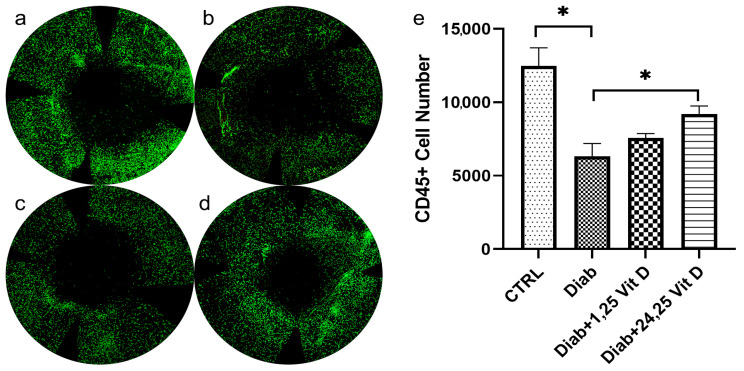
Topical 24,25 Vit D increases CD45+ cell numbers in the anterior stroma of diabetic mouse corneas. Mice were wounded with a 2 mm corneal epithelial debridement, and corneas were collected and immunostained with an anti-CD45 antibody (red) 8 h after corneal abrasion in (**a**) CTRL, (**b**) diabetic, (**c**) diabetic plus 1,25 Vit D and (**d**) diabetic plus 24,25-Vit D. (**e**) CD45+ numbers were quantified using the Imaris image processing software Spots package, demonstrating that 24,25 Vit D treatment elevated CD45+ immune cell numbers in wounded diabetic mouse corneas (*n* = 3, * *p* < 0.05).

## Data Availability

Data will be made available upon request from the corresponding author.

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
