# Peer review of "Effects of Topical 1,25 and 24,25 Vitamin D on Diabetic, Vitamin D Deficient and Vitamin D Receptor Knockout Mouse Corneal Wound Healing"

_biomolecules, 2023, doi:10.3390/biom13071065_

Round 1

Reviewer 1 Report

This is a well-written manuscript that builds on the evidence and prior literature on the role vitamin D has on corneal wound healing. The findings are compelling, with results presented in a succinct manner easily understandable to both clinicians and researchers alike. Only some minor concerns remain:

1.  The authors should briefly mention the methodology of their study in the abstract prior to demonstrating the results or findings of their study. 

2. For line 211 of the Results section, why does VDR inactivity preclude 1,25 D from being used. The authors should briefly explain this given that not all readers may be familiar with the biochemical processes underlying this rationale. This was briefly touched on in the second paragraph of the discussion, but should be lightly explained to provide some context to the reader why the decision is made to not use 1,25D but instead with 24,25 D in the Methods section, or following this line in the Results section.

3. Check syntax of the first line of the Conclusion: ‘Topical 1,25 Vit D significantly application accelerated corneal wound healing…’ needs to be amended.

4. For the conclusion, the authors should provide some insight as to what these findings could mean for the future of therapeutic management of corneal wound healing, instead of just recapitulating the findings of the study. For instance, is there potential for these topical applications to be effective for corneal injuries? Could the difference in immunological impact between the two topical vitamin D applications potentially translate to differences in clinical or therapeutic impact?

Author Response

We thank the reviewers for their thoughtful comments. We have addressed all of the reviewer comments below, with new additions to the manuscript added in green font.

Review 1

  1.  The authors should briefly mention the methodology of their study in the abstract prior to demonstrating the results or findings of their study. 

Response. Brief description has been added to the Abstract: For this study, wound healing and recruitment of CD45+ cells into the wound area of normoglycemic and diabetic mice were examined following corneal epithelial debridement and treatment with 1,25-dihyroxyvitamin D (1,25 Vit D) or 24,25-dihydroxyvitamin D (24,25 Vit D).

  1. For line 211 of the Results section, why does VDR inactivity preclude 1,25 D from being used. The authors should briefly explain this given that not all readers may be familiar with the biochemical processes underlying this rationale. This was briefly touched on in the second paragraph of the discussion, but should be lightly explained to provide some context to the reader why the decision is made to not use 1,25D but instead with 24,25 D in the Methods section, or following this line in the Results section.

Response. Changed the sentence to read “Because VDR was not active and 1,25 D is the primary agonist of VDR, VDR KO mice were not treated with 1,25 D.”

  1. Check syntax of the first line of the Conclusion: ‘Topical 1,25 Vit D significantly application accelerated corneal wound healing…’ needs to be amended.

Response. Edited to “Topical 1,25 Vit D application significantly accelerated…”

  1. For the conclusion, the authors should provide some insight as to what these findings could mean for the future of therapeutic management of corneal wound healing, instead of just recapitulating the findings of the study. For instance, is there potential for these topical applications to be effective for corneal injuries? Could the difference in immunological impact between the two topical vitamin D applications potentially translate to differences in clinical or therapeutic impact?

Response. We have added the following text to the end of the Conclusions section… These results indicate that topical 1,25 Vit D, and potentially topical 24,24 Vit D, may be useful for treating cornea epithelial wounds, particularly in Vit D deficient and diabetic patients. Reduced CD45+ cell recruitment in diabetic and VDR KO mouse corneas, which typically have worse epithelial wound healing outcomes compared to normal corneas, would indicate that CD45+ cell recruitment, at least in moderation, is beneficial for epithelial wound healing. Given the increased CD45+ recruitment in wounded diabetic corneas treated with 24,25 Vit D, this might be a preferable treatment for wounded diabetic corneas. The short half-life of 1,25 D and high cost of both 1,25 and 24,25 D3 may make cholecalciferol (Vit D3), which can be converted to both 1,25 D3 and 24,25 D3 in the cornea1-3, a more reasonable choice for topical treatment of corneal wounds in diabetic and vitamin D deficient patients.

Reviewer 2 Report

In many situations, delayed or prolonged corneal wound healing and non-healing wounds can lead to discomfort or visual loss. The authors show here that topical 1,25-dihyroxyvitamin D (1,25 Vit D) significantly increased corneal wound healing rates of normoglycemic, diabetic, and diabetic Vit D deficient mice. 24,25-dihydroxyvitamin D (24,25 Vit D) significantly increased corneal wound healing of diabetic Vit D deficient and Vit D receptor knockout (VDR KO) mice. Interestingly, CD45+ cell numbers were reduced in diabetic and VDR KO mouse corneas compared to normoglycemic mice, and 24,25 Vit D increased the recruitment of CD45+ cells to diabetic mouse corneas after epithelial debridement. The data show that topical Vit D promotes corneal wound healing impaired in diabetes and supports previous work that the Vit D corneal wound healing effect is not totally VDR-dependent. The paper is well presented, contains original and well described data, and is within the scope of the journal.

This reviewer has the following concerns about the manuscript.

1. Please specify the duration of diabetes before the wound healing experiments.

2. Please provide details on the generation of the Vit D deficient mice.

3. The difference between Fig. 8 a and b seems to be much higher than the graph shows. Please revisit. Likewise, in Fig. 9 a, b, the difference between diabetic and non-diabetic corneas does not seem significant contrary to the graph. Also, panel d seems to have more cells than panel a, again, contrary to the graph.

4. How long would the immune cells persist after the wounding is complete?

5. The Introduction would benefit from a better description of wound healing abnormalities in diabetic corneas mentioning molecular mechanisms and potential players.

6. Please acknowledge as a limitation that the immune cells were not typed in this work. It could be interesting as diabetes might change the proportions of immune populations in the cornea.

Author Response

We thank the reviewers for their thoughtful comments. We have addressed all of the reviewer comments below, with new additions to the manuscript added in green font.

Review 2

 Please specify the duration of diabetes before the wound healing experiments.

Response.  We added that diabetes duration was 10 weeks to lines 110 and 111.

  1. Please provide details on the generation of the Vit D deficient mice.

Response. We have modified the description of generating VDD mice in the methods section (line 102): As previously described and characterized4, VDD was induced by feeding mice a Vit D deficient diet (TD.89123 Diet; Envigo) for 4 weeks. VDD control mice were fed TD.89124 vitamin D control diet (Envigo).

  1. The difference between Fig. 8 a and b seems to be much higher than the graph shows. Please revisit. Likewise, in Fig. 9 a, b, the difference between diabetic and non-diabetic corneas does not seem significant contrary to the graph. Also, panel d seems to have more cells than panel a, again, contrary to the graph.

Response. We have changed the figures in question to more effectively represent the data in the graphs.

  1. How long would the immune cells persist after the wounding is complete?

Response. It has previously been shown that neutrophils persist for about 48 hours after epithelial wounding. This has now been included in lines 351-352: Neutrophil recruitment begins within 15 min and reaches a plateau level by 4–8 h in mice 5, with neutrophil numbers returning to baseline levels around 48 h5.

  1. The Introduction would benefit from a better description of wound healing abnormalities in diabetic corneas mentioning molecular mechanisms and potential players.

Response. We have added to lines 32-41 of the Introduction: Diabetic keratopathy is the most frequent clinical condition affecting the cornea in diabetics, occurring in approximately 70% of all diabetics 6-8. Diabetic keratopathy can exhibit several clinical manifestations, including (but not limited to) persistent corneal epithelial erosion, delayed epithelial regeneration, an altered ocular surface immune response, increased corneal thickness, altered stromal crosslinking and biomechanical properties, and decreased corneal sensitivity 8, 9. These features are typically caused by changes within the corneal epithelial basement membrane, deposition of glycation products, corneal nerve ending damage, reduced tear secretion, and oxidative stress related to the hyperglycemic condition 8.

  1. Please acknowledge as a limitation that the immune cells were not typed in this work. It could be interesting as diabetes might change the proportions of immune populations in the cornea.

Response. We have added “A limitation to the current study is that immune cells were only immunolabeled with an anti-CD45 antibody, and not immunophenotyped by flow cytometry.” To lines 402-404.

Round 2

Reviewer 2 Report

The authors have alleviated all concerns. The only recommended change in the Introduction would be that up to 70% of diabetics suffer from corneal abnormalities (40-70% by different accounts)